# Concurrent Mutations in *SF3B1* and *PHF6* in Myeloid Neoplasms

**DOI:** 10.3390/biology12010013

**Published:** 2022-12-21

**Authors:** Zhuang Zuo, L. Jeffrey Medeiros, Sofia Garces, Mark J. Routbort, Chi Young Ok, Sanam Loghavi, Rashmi Kanagal-Shamanna, Fatima Zahra Jelloul, Guillermo Garcia-Manero, Kelly S. Chien, Keyur P. Patel, Rajyalakshmi Luthra, C. Cameron Yin

**Affiliations:** 1Department of Hematopathology, The University of Texas MD Anderson Cancer Center, Houston, TX 77030, USA; 2Department of Leukemia, The University of Texas MD Anderson Cancer Center, Houston, TX 77030, USA

**Keywords:** *SF3B1*, *PHF6*, myeloid neoplasms

## Abstract

**Simple Summary:**

The advent of next-generation sequencing has elucidated the understanding of the genetic landscape of myeloid neoplasms. Most myeloid neoplasms carry more than one gene mutation at their initial presentation or develop them during the disease course. The patterns of and interactions between these mutated genes are of great research interest and may improve our ability to diagnose and prognosticate patients. It is known that certain gene mutations have a cooperative effect in the pathogenesis of myeloid neoplasms, whereas other gene mutations are mutually exclusive. A commonly cited example of mutually exclusive mutations is *SF3B1* and *PHF6*. This observation, however, has never been rigorously assessed. Since *SF3B1* and *PHF6* mutations can both serve as drivers of mutations and play key roles in the development of myeloid neoplasms, it is of clinical importance to clarify this issue. We report the clinicopathologic and molecular features of 21 myeloid neoplasms with double *SF3B1* and *PHF6* mutations. We summarize that concurrent mutations in *SF3B1* and *PHF6* are rare, but they do exist in a variety of myeloid neoplasms. *SF3B1* mutations usually occur as early initiating events whereas *PHF6* mutations occur late, with a role in disease progression. The presence of a larger number of other co-mutated genes suggests that the neoplastic cells have gone through active clonal evolution.

**Abstract:**

It has been reported that gene mutations in *SF3B1* and *PHF6* are mutually exclusive. However, this observation has never been rigorously assessed. We report the clinicopathologic and molecular genetic features of 21 cases of myeloid neoplasms with double mutations in *SF3B1* and *PHF6*, including 9 (43%) with myelodysplastic syndrome, 5 (24%) with acute myeloid leukemia, 4 (19%) with myeloproliferative neoplasms, and 3 (14%) with myelodysplastic/myeloproliferative neoplasms. Multilineage dysplasia with ring sideroblasts, increased blasts, and myelofibrosis are common morphologic findings. All cases but one had diploid or non-complex karyotypes. *SF3B1* mutations were detected in the first analysis of all the patients. *PHF6* mutations occurred either concurrently with *SF3B1* mutations or in subsequent follow-up samples and are associated with disease progression and impending death in most cases. Most cases had co-mutations, the most common being *ASXL1*, *RUNX1*, *TET2*, and *NRAS*. With a median follow-up of 39 months (range, 3-155), 17 (81%) patients died, 3 were in complete remission, and 1 had persistent myelodysplastic syndrome. The median overall survival was 51 months. In summary, concurrent mutations in *SF3B1* and *PHF6* are rare, but they do exist in a variety of myeloid neoplasms, with roles as early initiating events and in disease progression, respectively.

## 1. Introduction

With the advent of next-generation sequencing (NGS) technologies, the genetic landscape of myeloid neoplasms has become increasingly elucidated. Gene mutations in myeloid neoplasms affect a number of cellular pathways, including epigenetic modification (*ASXL1*, *DNMT3A*, *EZH2*, *IDH1*, *IDH2*, *TET2*), RNA splicing (*SF3B1*, *SRSF2*, *U2AF1*, *ZRSR2*), transcriptional regulation (*CEBPA*, *ETV6*, *PHF6*, *RUNX1*, *TP53*, *WT1*), and proliferative signaling pathways (*FLT3*, *JAK2*, *KRAS*, *NRAS*). It is also known that certain combinations of gene mutations have a cooperative effect in the pathogenesis of myeloid neoplasms, whereas other gene mutations are thought to be mutually exclusive [1,2,3,4].

Splicing factor 3b subunit 1 (*SF3B1*) is one of several genes involved in RNA splicing that is mutated in a variety of myeloid neoplasms. *SF3B1* mutations occur most commonly in myelodysplastic syndromes (MDS) and are best known for their association with ring sideroblasts (RS), which are signature phenotypes [5]. PHD finger protein 6 (*PHF6*) is involved in transcriptional regulation and the wild-type protein localizes to the nucleolus, associates with ribosomal RNA (rRNA) promoters, and suppresses rRNA transcription. Mutations of *PHF6* have been described in congenital syndromes and hematopoietic neoplasms [6,7].

A commonly cited example in the literature of mutually exclusive gene mutations is *SF3B1* and *PHF6* [6]. This observation, however, has never been rigorously assessed, and data from a recent study showed both *SF3B1* and *PHF6* mutations in 2 of 75 (2.7%) cases of acute myeloid leukemia (AML) [7]. From this study, it is unclear whether these mutations were detected simultaneously in the same sample or sequentially in different samples from the same patient. Since *SF3B1* and *PHF6* mutations can both serve as driver mutations and play key roles in the development of a variety of myeloid neoplasms, it is of clinical importance to clarify the issue of their potential mutual exclusivity. The aim of this study is to report a series of myeloid neoplasms in which concurrent *SF3B1* and *PHF6* mutations were detected, along with their clinicopathologic and molecular features.

## 2. Materials and Methods

### 2.1. Study Group

Following institutional review board approval, we retrospectively screened myeloid neoplasms that were assessed using a targeted 81-gene NGS panel test [8] performed as a part of the routine workup at our institution from January 1, 2017 through September 30, 2021. Clinical and laboratory data were obtained by review of the medical records. The study was conducted in accord with the Declaration of Helsinki.

### 2.2. Morphologic Evaluation

We reviewed hematoxylin-eosin stained core biopsy and clot specimens as well as Wright-Giemsa stained peripheral blood (PB) smears and bone marrow (BM) aspirate smears/touch imprints. Differential counts of 200- and 500-cells were performed manually on PB and BM smears, respectively. Cytochemical stains for iron and histochemical stains for reticulin and collagen were performed using standard methods.

### 2.3. Cytogenetic Analysis

Cytogenetic analysis was performed on metaphase cells prepared from BM aspirate specimens cultured for 24–48 h without mitogens, using standard techniques. Giemsa-banded metaphases were analyzed, and the results were reported using the International System for Human Cytogenetic Nomenclature, 2020.

### 2.4. Next-generation Sequencing

Genomic DNA was PCR-amplified and subjected to mutation analysis using a panel of 81 genes that commonly mutated in hematopoietic neoplasms on an Illumina MiSeq NGS platform (Illumina Inc., San Diego, CA, USA), as described previously [8]. The panel of 81 genes included *ANKRD26*, *ASXL1*, *ASXL2*, *BCOR*, *BCORL1*, *BRAF*, *BRINP3*, *CALR*, *CBL*, *CBLB*, *CBLC*, *CEBPA*, *CREBBP*, *CRLF2*, *CSF3R*, *CUX1*, *DDX41*, *DNMT3A*, *EED*, *ELANE*, *ETNK1*, *ETV6*, *EZH2*, *FBXW7*, *FLT3*, *GATA1*, *GATA2*, *GFI1*, *GNAS*, *HNRNPK*, *HRAS*, *IDH1*, *IDH2*, *IKZF1*, *IL2RG*, *IL7R*, *JAK1*, *JAK2*, *JAK3*, *KDM6A*, *KIT*, *KMT2A*, *KRAS*, *MAP2K1*, *MPL*, *NF1*, *NOTCH1*, *NPM1*, *NRAS*, *PAX5*, *PHF6*, *PIGA*, *PML*, *PRPF40B*, *PTEN*, *PTPN11*, *RAD21*, *RARA*, *RUNX1*, *SETBP1*, *SF1*, *SF3A1*, *SF3B1*, *SH2B3*, *SMC1A*, *SMC3, SRSF2*, *STAG1*, *STAG2*, *STAT3*, *STAT5A*, *STAT5B*, *SUZ12*, *TERC*, *TERT*, *TET2*, *TP53, U2AF1*, *U2AF2*, *WT1*, and *ZRSR2.*

## 3. Results

### 3.1. Concurrent SF3B1 and PHF6 Mutations Occur at Low Frequency in Myeloid Neoplasms

Among a total of 6478 cases identified within the study period, *SF3B1* mutations were detected in 562 patients and *PHF6* mutations were detected in 245 patients, accounting for 8.7% and 3.8% of all patients, respectively. Most of these patients had been tested multiple times during diagnosis and/or follow-up visits (median = 4; range 1–11). All test results were retrieved for analysis.

Sixty-seven distinct *SF3B1* mutations were detected, most of which were located in the H5 and H6 of the so-called HEAT (Huntingtin, elongation factor 3, protein phosphatase 2A, and the yeast PI3-kinase TOR1) motifs [9]. Over 98% of these mutations were missense mutations, and nearly half (49%) were K700E mutation. Other common mutations occurred at codon K666 (22%; 65% of which were K666N), codon R625 (7%), and codon H662 (5%).

A total of 198 distinct mutations in *PHF6* were detected. These mutations were widely spanned across the whole coding region but were relatively more concentrated in the two PHD-like zinc finger domains [6]. The most frequent mutations included R274 (R274Q and R274*) in 8%, I314 in 6%, R225* in 4%, R116* in 4%, C297 in 4%, and R342* in 3% of myeloid neoplasms. About 28% of these mutations were nonsense mutations, likely resulting in the inactivation of this tumor suppressor gene.

We identified 23 neoplasms that had mutations in both *SF3B1* and *PHF6*, representing 0.4% of all neoplasms assessed. Among all the cases with *SF3B1* mutation in this study, these 23 cases accounted for 4%, compared with other frequently mutated genes in the *SF3B1*-mutated group, such as *TET2* (18%), *DNMT3A* (17%), *JAK2* (16%), *ASXL1* (13%), and *TP53* (13%). Among all the cases with *PHF6* mutations, these 23 patients accounted for 9%, compared with other mutated genes in the *PHF6*-mutated group, such as *ASXL1* (40%), *TET2* (34%), *RUNX1* (25%), *DNMT3A* (19%), *NRAS* (17%), *TP53* (16%), and *JAK2* (14%). Further analysis revealed that, in two patients the *SF3B1* and *PHF6* mutations were never detected simultaneously in the same sample, suggesting that mutations of these two genes likely did not arise from the same clone. For that reason, these two cases were excluded, leaving a study cohort of 21 cases that account for 0.3% of all myeloid neoplasms assessed during the study period.

### 3.2. Concurrent SF3B1 and PHF6 Mutations Occur in a Variety of Myeloid Neoplasms

The study group included 15 men and 6 women with a median age of 69 years (range, 51–84). The pathologic diagnoses encompassed the full spectrum of myeloid neoplasms and included 9 (43%) cases with MDS, 5 (24%) with AML, 4 (19%) with myeloproliferative neoplasms (MPN), and 3 (14%) with MDS/MPN. All (100%) of the MDS cases had increased RS and 4 (44%) had excess blasts (≥5%, MDS-EB). Most AML patients (3/5, 60%) had AML with myelodysplasia-related changes (AML-MRC), one had AML with maturation, and one had AML with minimal differentiation. All three (100%) patients with MPN had primary myelofibrosis (PMF); one patient did not have a BM biopsy done at our hospital. One patient with MDS/MPN presented in the accelerated phase, and the other two came to us after chemotherapy. The diagnosis and clinicopathologic features are summarized in Table 1.

### 3.3. Morphologic Findings of Myeloid Neoplasms with SF3B1 and PHF6 Mutations

The BM biopsies were hypercellular (median cellularity, 80%). Dysplasia was common in all disease entities, with all MDS/MPN (3/3, 100%) and most MDS (8/9, 89%), AML (3/4, 75%), and MPN (2/3, 67%) having dysplasia involving ≥2 lineages. Increased RS (≥5%) were noted in all cases assessed except one PMF case. Apart from AML, increased blasts (≥5%) were detected in 2/3 (67%) of MDS/MPN, 4/9 (44%) of MDS, and 1/3 (33%) of MPN cases. Leukemic transformation occurred in 6/9 (67%) of MDS and 2/3 (67%) of MDS/MPN cases, with a median interval of 37 months from the initial diagnosis to the development of leukemia (range, 2-141). Myelofibrosis was also a common finding, seen in 3/3 (100%) of MPN, 3/3 (100%) of MDS/MPN, 5/6 (83%) of MDS, and 3/4 (75%) of AML cases. The morphologic findings from representative cases are illustrated in Figure 1.

### 3.4. Cytogenetic Findings of Myeloid Neoplasms with SF3B1 and PHF6 Mutations

Among 20 patients with cytogenetic data available, 12 (60%) had a normal karyotype, 7 (40%) had simple abnormal karyotypes with 1–2 chromosomal aberration(s), and 1 (10%, AML) had a complex karyotype with 3 chromosomal aberrations. No specific recurrent karyotypic abnormalities were identified, except for del20q, which was detected in three (15%) cases (2 PMF and 1 MDS/MPN) (Table 1).

### 3.5. Molecular Findings of Myeloid Neoplasms with SF3B1 and PHF6 Mutations

In nine patients, NGS was performed at the time of the initial diagnosis or within 3 months of the diagnosis, and *SF3B1* mutations were present in all of these nine (100%) patients. In the remaining 12 patients, NGS was performed 11–126 months (median, 12) after the initial diagnosis and *SF3B1* mutations were detected at first test in all 12 (100%) patients. Overall, *SF3B1* mutations were detected in the first analysis of all 21 (100%) patients. *PHF6* mutations occurred either concurrently with *SF3B1* mutations (n = 14, 67%) or in subsequent follow-up samples (n = 7, 33%, median interval between the two mutations, 10 months; range, 5–21 months). In the latter scenario, the emergence of *PHF6* mutations was accompanied by an increased blast count in two MDS/MPN cases and one MDS case, leukemia transformation in one MDS case, and clonal cytogenetic evolution with +8 in one MPN case. Moreover, the appearance of *PHF6* mutations was associated with impending death in several cases (two MDS/MPN, one MDS, and one MPN), with these patients succumbing to the disease shortly after the emergence of *PHF6* mutations.

All *SF3B1* mutations were missense mutations, with the hotspots at codon K700 (n = 8) and K666 (n = 7). One MDS case had two *SF3B1* mutations. *PHF6* mutations consisted of 16 missense mutations, 7 nonsense mutations, 7 frameshifts, and 1 splice mutation, without hotspots. Four cases had more than one *PHF6* mutations. The distribution of *SF3B1* and *PHF6* mutations is illustrated in Figure 2A.

The VAF of *SF3B1* mutations ranged from 2–45% (median, 32%) with 18 (86%) cases having a VAF of ≥5%. The VAF of *PHF6* mutations ranged from 1–85% (median, 13%), with 14 (67%) cases having a VAF of ≥5%. In 12 cases, the VAF levels of both mutations were ≥5%, and both mutations were maintained at substantial levels in these cases until the last follow-up.

We also evaluated the frequencies of other coexisting mutations in these cases (Figure 2B). In addition to *SF3B1* and *PHF6*, all but one PMF case had at least one other gene mutation. The median number of mutated genes in this cohort was 7, higher than that of the *SF3B1*-mutated group (median = 3.5) or the *PHF6*-mutated group (median = 5). The most common coexisting mutations affected *TET2* (n = 8), *ASXL1* (n = 7), *DNMT3A* (n = 7), and *RUNX1* (n = 7). However, in three cases (one MDS, one AML, and one PMF), mutations of *PHF6* and *SF3B1* had nearly equal VAFs and hence were likely in the same dominant clone; there were also no or fewer (≤2) other genes involved, with zero in PMF cases, one in an MDS case, and two in AML cases (Figure 2B). *TP53* mutations were detected in three (14%) cases, and the mutations were either with low VAFs (<2%) and transient (in an MDS case and an MDS/MPN case) or persistent at low levels (VAF<3%, in an MDS case), suggesting that *TP53* may not be a key player in these cases. The VAFs of all mutations from each of these cases are provided in Appendix A. Mutation clone sizes in a representative test of each of the 12 myeloid neoplasms cases with a significant level of concurrent *SF3B1* and *PHF6* mutations are plotted in Figure 2C.

### 3.6. Clinical Findings of Myeloidoutcomes of Myeloid Neoplasms with SF3B1 and PHF6 Mutations

All patients received chemotherapy, and four MDS patients additionally received allogeneic stem cell transplantation (ASCT). With a median follow-up of 39 months (range, 3–155), the clinical outcomes of these patients were poor. Seventeen (81%) patients died by the end of the study interval, including two of the four MDS patients who underwent transplantation. Three were in complete remission, including two MDS patients after transplantation and one AML patient after chemotherapy. One patient had persistent MDS (Table 1). Most patients, except for four MDS (after transplantation) and four AML patients, never achieved complete remission. Two MDS patients and one AML patient shortly relapsed. The median overall survival (OS) of these patients was 51 months.

## 4. Discussion

In recent years, large-scale analysis using NGS has become feasible in clinical practice and a large amount of data on gene mutation patterns in myeloid neoplasms has accumulated. These data have shown that most myeloid neoplasms have more than one gene mutation [1,2,3,10,11]. The patterns of and interactions between these mutated genes is of great research interest and may improve our ability to diagnose and prognosticate patients with myeloid neoplasms [4,12].

In this study, we analyzed the NGS panel test results of 6478 patients to address the question of whether *SF3B1* and *PHF6* mutations are mutually exclusive in myeloid neoplasms. Our findings suggest that these double-mutated cases are uncommon (<1%), but far from mutually exclusive, in contrast with the literature [6]. In this study, these cases account for roughly 4% of all cases with *SF3B1* mutations or 9% of all cases with *PHF6* mutations. These neoplasms also usually harbor a larger number of other co-mutated genes, suggesting that the neoplastic cells have gone through active clonal evolution. The disease distribution of these cases was generally consistent with the literature’s reported frequencies in cases with *SF3B1* [5] or *PHF6* [6,7] mutations separately, except that MPN cases seemed slightly more frequent. One possible explanation for myeloid neoplasms with both *SF3B1* and *PHF6* mutations not having been reported in the literature may be the lack of large-scale studies of myeloid neoplasms that have been assessed for *PHF6* mutations.

*SF3B1* is one of the most commonly mutated genes in MDS. *SF3B1* mutations are particularly common (>80%) in cases of MDS with RS and confer a favorable prognosis [13]. *SF3B1* mutations have also been reported in approximately 20% of patients with MDS with isolated del(5q), without any significant impact on their overall survival [14]. In AML with RS, on the other hand, *SF3B1* mutations have been associated with an inferior clinical outcome [5]. *SF3B1* mutations have also been reported in about 10% of *BCR-ABL1*–negative MPNs [13], but its prognostic significance is still unclear, with contradictory reports. Some researchers reported a role of *SF3B1* mutations in myelofibrotic progression in patients with polycythemia vera and essential thrombocythemia [15], whereas others concluded that *SF3B1* mutations did not correlate with fibrotic evolution [16]. *SF3B1* mutations are disease-defining in MDS/MPN with RS and thrombocytosis (MDS/MPN-RS-T) and are seen in about 5% of chronic myelomonocytic leukemia (CMML), without a well-established association with leukemic transformation and overall survival [17]. Moreover, *SF3B1* mutations also occur frequently in non-myeloid neoplasms such as chronic lymphocytic leukemia/small lymphocytic lymphoma, and have been associated with resistance to fludarabine therapy and poor clinical outcomes [18]. It has been reported that an *SF3B1* mutation is an early event in myeloid neoplasms, and in MDS/MPN, *SF3B1* mutations seem to represent founder mutations followed by secondary hits in genes involved in other signaling pathways, such as *JAK2* [17]. In keeping with this, *SF3B1* mutations were detected in the first tests in all cases in this study. A recent study suggested that not all *SF3B1* mutations are equal. The most common *SF3B1* K700E mutation confers a favorable prognosis in MDS cases and is associated with a lower risk of progression to AML compared with patients with wild-type *SF3B1* [2,14], whereas a K666N mutation has been found to be associated with a poorer prognosis [19]. Only eight (38%) of the *SF3B1* mutations in this *SF3B1* and *PHF6* double-mutated cohort were K700E, and K666 mutation occurred in a similar percentage of patients (n = 7, 33%). All patients with *SF3B1* mutations at codon K666 died at the end of the follow-up period, though their OS varied largely. Moreover, we noticed that in cases with *PHF6* mutations at a substantial level and in dominant clones, the co-mutated *SF3B1* was more likely to be of the K666 type. This may lead to speculation that *SF3B1* K666 mutations could provide a preferred microenvironment for *PHF6* mutated clones to gain a competition edge in during clonal evolution.

Somatic *PHF6* mutations occur more frequently in T-lymphoblastic leukemia/lymphoma and have only been reported in approximately 3% of myeloid neoplasms [6,20], which is consistent with our findings in this study. The potential role of *PHF6* mutations in the pathogenesis of myeloid neoplasms is poorly understood, but *PHF6* appears to act as a tumor suppressor gene, regulating the transcription of signaling genes and the DNA damage response [7]. As myeloid neoplasms with *PHF6* mutations are rare, the clinical characteristics of patients with these myeloid neoplasms are much less established compared to those of patients with myeloid neoplasms with *SF3B1* mutations. The findings in the current study contribute more data to this entity. Myeloid neoplasms with *SF3B1* and *PHF6* double mutations tend to show morphologic evidence of myelodysplasia, increased reticulin fibrosis, an increased blast count, and the risk of leukemia transformation. The prognostic significance of *PHF6* mutations remains unclear. Some studies showed that *PHF6* mutations predicted a shorter overall survival in patients with intermediate-risk AML [20], whereas others showed that *PHF6* mutations had no prognostic impact when compared with patients with wild-type *PHF6* [5]. Noticeably, the clinical outcome of these double-mutated cases in this study was very poor, with a median survival more in keeping with that of patients with *PHF6*-mutated neoplasms than that of *SF3B1*-mutated neoplasms.

Another reported feature of *PHF6* mutations is that these mutations are associated with an increased number of other mutated genes (>3 besides *PHF6*), including *ASXL1*, *EZH2*, *RUNX1*, *TET2*, *U2AF1*, and *ZRSR2* [6], but are mutually exclusive with *SF3B1* [5,21]. The findings in this study are consistent with earlier reports and confirm that *PHF6* mutations are significantly associated with *ASXL1* and *RUNX1* mutations. However, in the double mutation cohort, we noticed that once both mutations reached dominant clone levels, there tended to be fewer other genes involved, suggesting that the combination of *SF3B1* and *PHF6* mutations may be sufficiently powerful to drive the disease’s progression. In addition, low frequencies and very low levels of *TP53* mutations were detected in these double-mutated cases. This finding could imply that mutations in *PHF6*, also acting as a tumor suppressor gene, may have been more important in the pathogenesis than *TP53* during clonal evolution. Moreover, *PHF6* mutations tend to occur late in the course of myeloid neoplasms, either concurrently with or subsequently to *SF3B1* mutations in all cases, and are associated with disease progression in a subset of the cases. All of these data suggest that *PHF6* mutations in myeloid neoplasms often arise late as a result of clonal evolution and that increasing levels of *PHF6* mutations are associated with poor clinical outcomes. It seems reasonable to suggest that the coexistence of these two mutations contributes to poor clinical outcomes.

A limitation of this study is that we did not perform single-cell sequencing and therefore we cannot prove with certainty that the *SF3B1* and *PHF6* mutations resided in the same clone. We are also aware of the fact that most of the cases in this study had other mutated genes, which could be confounding factors for the assessment of the clinical findings of this cohort. Nevertheless, the findings in this study show that myeloid neoplasms with *SF3B1* and *PHF6* mutations do occur uncommonly, and seem to be associated with some distinctive features.

## 5. Conclusions

In summary, concurrent mutations in the *SF3B1* and *PHF6* genes are rare in myeloid neoplasms, but they do exist. The simultaneous occurrence of both mutations most likely is the result of the active clonal evolution of the neoplastic cells with no specific associated disease type preference. It is possible that myeloid neoplasms with *SF3B1* and *PHF6* double mutations may represent a unique entity among myeloid neoplasms.

## Figures and Tables

**Figure 1 biology-12-00013-f001:**
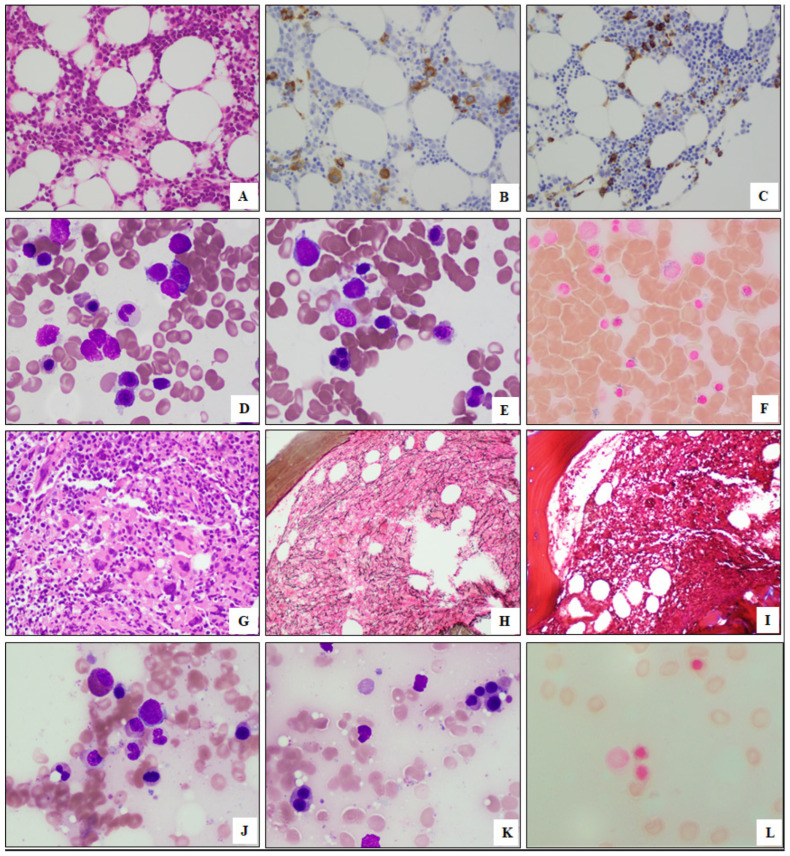
Morphologic features of representative cases of myeloid neoplasms with *SF3B1* and *PHF6* mutations. (**A**–**F**) A case of myelodysplastic syndrome with multilineage dysplasia, increased blasts, and fibrosis. The bone marrow is hypercellular for age (**A**,**H**&**E**) with dysmegakaryopoiesis (**B**, CD61) and increased blasts (**C**, CD34). The aspirate smear shows dysgranuopoiesis (**D**, Wright-Giemsa), dyserythropoiesis (**E**, Wright-Giemsa), and increased ring sideroblasts (**F**, Prussian Blue) (**A**–**C**, 200×; **D**–**F**, 500×). (**G**–**L**) A case of primary myelofibrosis. The bone marrow is hypercellular (**G**,**H**&**E**), with reticulin fibrosis (**H**), without collagen fibrosis (**I**). The aspirate smear shows dysgranulopoiesis (**J**, Wright-Giemsa), dyserythropoiesis (**K**, Wright-Giemsa), and increased ring sideroblasts (**L**, Prussian Blue) (**G**–**I**, 200×; **J**–**L**, 500×).

**Figure 2 biology-12-00013-f002:**
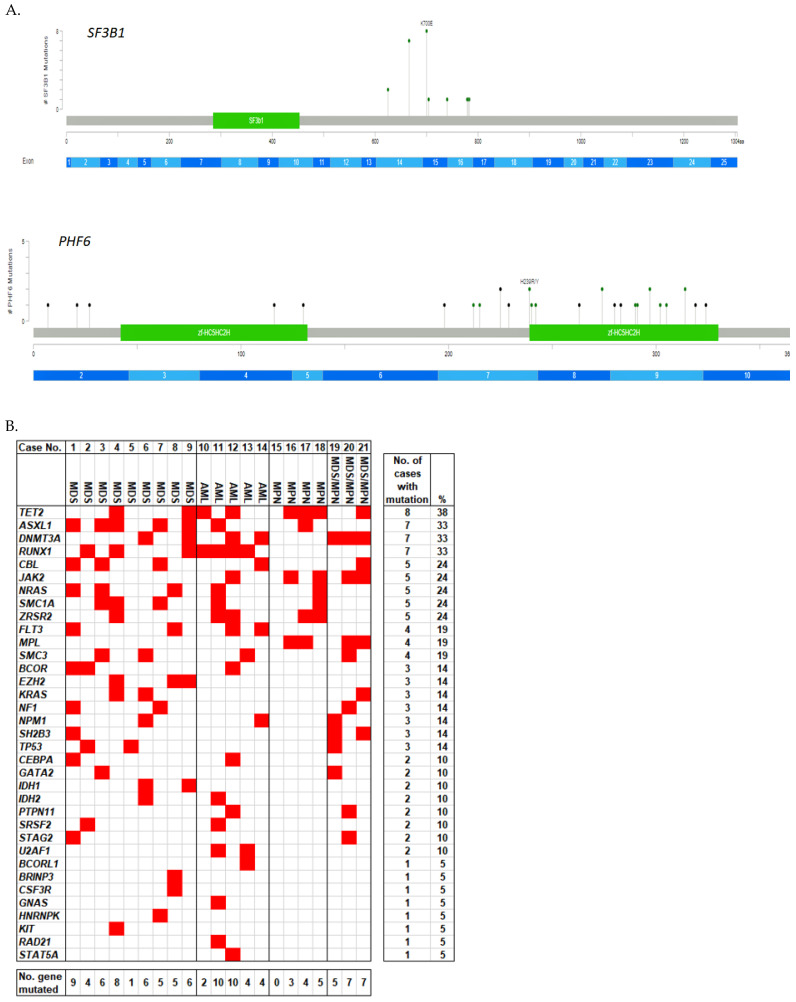
Molecular features of cases of myeloid neoplasms with *SF3B1* and *PHF6* mutations. (**A**) The location and number of *SF3B1* and *PHF6* mutations. (**B**) The mutation profile heatmap. The cases were divided into four categories: myelodysplastic syndrome (MDS), acute myeloid leukemia (AML), myeloproliferative neoplasms (MPN), and myelodysplastic/myeloproliferative neoplasms (MDS/MPN). Each column represents a single case. Overall, the most frequent concurrent mutations were *TET2*, *ASXL1*, *DNMT3A*, and *RUNX1*. (**C**) Mutation clone size of each case. Normalized variant allele frequencies (VAFs) in a representative test of each of the 12 myeloid neoplasms cases with a significant level of concurrent *SF3B1* and *PHF6* mutations are shown.

**Table 1 biology-12-00013-t001:** Clinicopathologic and molecular genetic features of 21 patients with myeloid neoplasms with *SF3B1* and *PHF6* mutations.

Case	Age	Sex	WBC	Hb	Platelet	LDH	Karyotype	*SF3B1*	*PHF6*	SCT	FU (m)	Outcome
MDS												
1	69	M	8.2	10.2	234	NA	inv(17)(p13q23) [20/20]	K666M	K21fs, R116*, F198fs, C242Y, G291E, C297R, C297Y	No	39	Died
2	51	M	8.9	12.1	116	204	del(11q)[9/20]	R625C, D781G	C305S, L324fs	Yes	14	CR
3	56	M	7.0	9.8	510	207	46,XY[20]	E783K	C212G	Yes	155	Died
4	77	M	2.0	8.6	244	130	46,XY[20]	S779P	C280fs	No	38	MDS
5	77	M	6.3	10.2	153	124	del(13)(q12q14) [3/19]	K666T	Q7*	No	65	Died
6	84	M	11.1	7.4	88	568	46,XY[20]	K666N	R319*	No	3	Died
7	74	M	1.7	9.8	41	159	add(1)(p36.1) [2/20]	R625C	H229fs, C283*	Yes	39	CR
8	62	M	7.2	9.9	336	441	46,XY[20]	K700E	R225*	Yes	71	Died
9	68	M	3.1	7.6	63	177	46,XY[20]	K700E	I290T	No	28	Died
AML												
10	70	F	6.9	6.9	5	135	46,XX[20]	K700E	c.585+1G>A p.?	No	11	Died
11	82	M	2.5	9.1	53	130	46,XY[20]	I704N	H302R	No	90	Died
12	82	M	1.2	11.3	111	232	–Y,+1, der(1;13)(q10;q10) [4/11]	K666N	R274Q	No	51	Died
13	75	M	1.0	8.0	52	189	46,XY[20]	K700E	K130fs	No	13	Died
14	58	F	5.8	8.9	199	126	46,XX[20]	K700E	H239R	No	9	CR
MPN												
15	67	F	2.9	8.6	39	247	del(20)(q11.2q13.3) [20]	K666N	H239Y	No	57	Died
16	59	M	7.1	9.0	56	375	del(20)(q11.2q13.3) [20]	K666T	I314T	No	37	Died
17	76	M	3.6	9.3	196	266	46,XY[20]	K666Q	R274Q	No	41	Died
18	58	M	NA	NA	NA	NA	NA	K700E	E27fs, R225*, Y240C	No	72	Died
MDS/MPN												
19	73	F	11.5	7.2	25	987	46,XX[20]	G740E	C215S	No	16	Died
20	69	F	17.2	5.2	595	666	46,XX[20]	K700E	F263*	No	87	Died
21	66	F	60.4	9.1	26	570	del(20)(q11.2q13.1) [6/20]	K700E	I314T	No	20	Died

AML—acute myeloid leukemia; CR —complete remission; F—female; FU (m)—follow-up (months); Hb—hemoglobin; LDH—lactate dehydrogenase; M—male; MDS—myelodysplastic syndrome; MDS/MPN—myelodysplastic/myeloproliferative neoplasms; MPN—myeloproliferative neoplasms; NA—not available; SCT—stem cell transplantation; WBC—white blood cells. Reference ranges: WBC, 4.0–11.0 K/uL; Hb, 14.0–18.0 g/dL for men and 12.0–16.0 g/dL for women; platelet, 140–440 K/uL; LDH, 135–225 IU/L.

## Data Availability

The data presented in this study are available on request from the corresponding authors. The data are not publicly available due to privacy.

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
