# Peer review of "Concurrent Mutations in SF3B1 and PHF6 in Myeloid Neoplasms"

_biology, 2022, doi:10.3390/biology12010013_

Round 1
Reviewer 1 Report
Concurrent Mutations in SF3B1 and PHF6 in Myeloid Neoplasms
Zuo and colleagues describe the clinicopathologic and molecular features of 21 cases with a myeloid neoplasms and concurrent SF3B1 and PHF6, a combination that is generally considered to be uncommon.
Comments
The manuscript contains some useful information and is mostly well written, but given the diverse diagnoses, features, outcomes and complexity of additional mutations there is no convincing evidence to support the suggestion that co-mutation of SF3B1 and PHF6 may represent a unique entity. This suggestion should be removed.
The variant allele fractions for each mutation should be given, and mutations should be presented in HGVS format (if necessary as a supplementary table) specifying the Refseq used. Cytogenetic findings should indicate the number of metaphases examined and the number that were positive for each abnormality.
PHF6 is on the X-chromosome and therefore the relationship between clone size and vaf is different for males and females. Fig 2c would be better as clone size rather than vaf on the y axis
Cases 1, 9 and 12 on Figure 2C have >1 mutation. This suggests selection for PHF6 mutations, and should be commented on in the text. Do these three cases have any other mutated genes in common?
Author Response
We very much appreciate the positive comments by the reviewers and here we provide a point-by-point response to each of the reviewers’ comments. We have highlighted the changes using the “Track Changes” function.
Reviewer 1
Zuo and colleagues describe the clinicopathologic and molecular features of 21 cases with a myeloid neoplasms and concurrent SF3B1 and PHF6, a combination that is generally considered to be uncommon.
Comments
The manuscript contains some useful information and is mostly well written, but given the diverse diagnoses, features, outcomes and complexity of additional mutations there is no convincing evidence to support the suggestion that co-mutation of SF3B1 and PHF6 may represent a unique entity. This suggestion should be removed.
Thanks! We agree and have removed this statement.
The variant allele fractions for each mutation should be given, and mutations should be presented in HGVS format (if necessary as a supplementary table) specifying the Refseq used. Cytogenetic findings should indicate the number of metaphases examined and the number that were positive for each abnormality.
We have provided a supplementary table to list the variant allele frequency of each mutation in HGVS format.
We have also provided the number of metaphases for each abnormality.
PHF6 is on the X-chromosome and therefore the relationship between clone size and vaf is different for males and females. Fig 2c would be better as clone size rather than vaf on the y axis.
This is a very good point. We have revised Fig 2c.
Cases 1, 9 and 12 on Figure 2C have >1 mutation. This suggests selection for PHF6 mutations, and should be commented on in the text. Do these three cases have any other mutated genes in common?
Multiple PHF6 mutations may be caused by stress. No unique co-mutation pattern could be found. The details are provided in Suppl table 1.
Reviewer 2 Report
GENERAL COMMENTS
This is a manuscript describing the clinical and laboratory findings of 21 patients with various myeloid malignancies, exhibiting SF3B1 and PHF6 mutations, identified either concurrently or sequentially. The authors provide potential interpretation of the hierarchical model of occurrence of these two mutations and discuss their pathophysiological consequences in the course of the disease. In view of the ongoing bulk of knowledge on the genetical / mutational models of disease pathogenesis, particularly in the field of MDS, this manuscript offers some important pieces of new information and speculations and deserves to be published. The quality of Tables and Figures is good. The language is good without any significant issue. I can only see some minor issues, mainly some missing pieces of information, which the authors should address and reply, before the manuscript could be accepted for publication. Please, see below my specific recommendations.
MINOR ISSUES
Simple summary, lines 16-17: Please rephrase as follows: “Most myeloid neoplasms carry at initial presentation or develop in the disease course more than one gene mutation.”
Results, lines 108-109: Please, provide the median number and the range of times, each patient has been tested with NGS.
Results, lines 135-144: Although the type of SF3B1 or PHF6 mutation is shown on Table I, the authors have to collectively describe in the text the most common type of mutation identified in each patient subgroup.
Results, lines 192-196: Again here, the reader would seek to see the distribution of the type of SF3B1 mutations (missence, nonsense, frameshift etc), according to each patient group (MDS, AML, MPN, MDS/MPN). Please, incorporate this information.
Discussion, line 239: Please, refer better to patients, not to “cases”.
Discussion, lines 251-254: Please, add a comment for the significance of the presence of SF3B1 mutations in patients with MDS and Del(5)q.
Discussion, lines 251-265: The authors need to report and discuss in short, at the discussion part, in which other non-myeloid (i.e. in CLL) and/or non-hematological malignancies SF3B1 mutations have been identified and which is their prognostic significance in the same way as they report it for PHF6 mutations.
Discussion, lines 271-274: Can you provide a potential explanation or at least speculation for this observation, regarding the K666 mutations?
Discussion, line 308: Since data on synergy of these two mutations are not yet available, I suggest to substitute the word “synergy” with the word “coexistence”.
Author Response
Dear Editor,
Attached is our revised manuscript “Concurrent Mutations in SF3B1 and PHF6 in Myeloid Neoplasms” (biology-2072813).
We very much appreciate the positive comments by the reviewers and here we provide a point-by-point response to each of the reviewers’ comments. We have highlighted the changes using the “Track Changes” function.
Reviewer 2
This is a manuscript describing the clinical and laboratory findings of 21 patients with various myeloid malignancies, exhibiting SF3B1 and PHF6 mutations, identified either concurrently or sequentially. The authors provide potential interpretation of the hierarchical model of occurrence of these two mutations and discuss their pathophysiological consequences in the course of the disease. In view of the ongoing bulk of knowledge on the genetical / mutational models of disease pathogenesis, particularly in the field of MDS, this manuscript offers some important pieces of new information and speculations and deserves to be published. The quality of Tables and Figures is good. The language is good without any significant issue. I can only see some minor issues, mainly some missing pieces of information, which the authors should address and reply, before the manuscript could be accepted for publication. Please, see below my specific recommendations.
MINOR ISSUES
Simple summary, lines 16-17: Please rephrase as follows: “Most myeloid neoplasms carry at initial presentation or develop in the disease course more than one gene mutation.”
We have made this change.
Results, lines 108-109: Please, provide the median number and the range of times, each patient has been tested with NGS.
We have provided this information.
Results, lines 135-144: Although the type of SF3B1 or PHF6 mutation is shown on Table I, the authors have to collectively describe in the text the most common type of mutation identified in each patient subgroup.
This information was provided in Results, lines 193-197.
Results, lines 192-196: Again here, the reader would seek to see the distribution of the type of SF3B1 mutations (missence, nonsense, frameshift etc), according to each patient group (MDS, AML, MPN, MDS/MPN). Please, incorporate this information.
Thanks for the suggestion! Since there is only a limited number of cases in each disease entities, and there is a wide spectrum of types of mutations, especially for PHF6, in each subgroup of diseases, we think that this information is better summarized in Table only, without duplication description.
Discussion, line 239: Please, refer better to patients, not to “cases”.
Changed.
Discussion, lines 251-254: Please, add a comment for the significance of the presence of SF3B1 mutations in patients with MDS and Del(5)q.
We have added a comment as requested.
Discussion, lines 251-265: The authors need to report and discuss in short, at the discussion part, in which other non-myeloid (i.e. in CLL) and/or non-hematological malignancies SF3B1 mutations have been identified and which is their prognostic significance in the same way as they report it for PHF6 mutations.
We have added a comment as requested.
Discussion, lines 271-274: Can you provide a potential explanation or at least speculation for this observation, regarding the K666 mutations?
We have provided a speculation.
Discussion, line 308: Since data on synergy of these two mutations are not yet available, I suggest to substitute the word “synergy” with the word “coexistence”.
Changed.